# DNA methylome combined with chromosome cluster-oriented analysis provides an early signature for cutaneous melanoma aggressiveness

Arnaud Carrier[1,2], Cécile Desjobert[1], Loic Ponger[3], Laurence Lamant[4], Matias Bustos[5], Jorge Torres-Ferreira[6,7], Rui Henrique[6,7,8], Carmen Jeronimo[6,7,8], Luisa Lanfrancone[9], Audrey Delmas[4], Gilles Favre[4], Antoine Daunay[10], Florence Busato[11], Dave SB Hoon[5], Jorg Tost[11], Chantal Etievant[1], Joëlle Riond[1,4], Paola B Arimondo[1,12]*

[1]Unité de Service et de Recherche USR 3388, CNRS-Pierre Fabre, Epigenetic Targeting of Cancer (ETaC), Toulouse, France; [2]Cancer Epigenetics Group, Josep Carreras Leukemia Research Institute (IJC), Barcelona, Spain; [3]CNRS UMR 7196, INSERM U1154, Sorbone university- National museum of natural history (NMNH), Paris, France; [4]Cancer Research Center of Toulouse, UMR 1037, INSERM, Université Toulouse III Paul Sabatier, Toulouse, France; [5]Department of Translational Molecular Medicine, Saint John's Cancer Institute, Providence Saint John's Health Center, Santa Monica, United States; [6]Cancer Biology and Epigenetics Group, Research Center (CI-IPOP)/P.CCC Porto Comprehensive Cancer Center, Portuguese Oncology Institute of Porto (IPO Porto), Porto, Portugal; [7]Department of Pathology, Portuguese Oncology Institute of Porto (IPO Porto)/P.CCC Porto Comprehensive Cancer Center, Porto, Portugal; [8]Department of Pathology and Molecular Immunology, Biomedical Sciences Institute (ICBAS), University of Porto, Porto, Portugal; [9]Department of Experimental Oncology, Instituto Europeo di Oncologia, Milan, Italy; [10]Laboratory for Functional Genomics, Fondation Jean Dausset-CEPH, Paris, France; [11]Laboratory for Epigenetics and Environment, Centre National de Recherche en Génomique Humaine, CEA-Institut de Biologie François Jacob, Evry, France; [12]EpiCBio, Epigenetic Chemical Biology, Department Structural Biology and Chemistry, Institut Pasteur, CNRS UMR 3523, Paris, France

*For correspondence:
paola.arimondo@cnrs.fr

Competing interest: The authors declare that no competing interests exist.

**Abstract** Aberrant DNA methylation is a well-known feature of tumours and has been associated with metastatic melanoma. However, since melanoma cells are highly heterogeneous, it has been challenging to use affected genes to predict tumour aggressiveness, metastatic evolution, and patients' outcomes. We hypothesized that common aggressive hypermethylation signatures should emerge early in tumorigenesis and should be shared in aggressive cells, independent of the physiological context under which this trait arises. We compared paired melanoma cell lines with the following properties: (i) each pair comprises one aggressive counterpart and its parental cell line and (ii) the aggressive cell lines were each obtained from different host and their environment (human, rat, and mouse), though starting from the same parent cell line. Next, we developed a multi-step genomic pipeline that combines the DNA methylome profile with a chromosome cluster-oriented analysis. A total of 229 differentially hypermethylated genes was commonly found in the aggressive cell lines. Genome localization analysis revealed hypermethylation peaks and clusters, identifying

eight hypermethylated gene promoters for validation in tissues from melanoma patients. Five Cytosine-phosphate-Guanine (CpGs) identified in primary melanoma tissues were transformed into a DNA methylation score that can predict survival (log-rank test, p=0.0008). This strategy is potentially universally applicable to other diseases involving DNA methylation alterations.

## Editor's evaluation

Predicting if a tumour has aggressive or metastatic characteristics would be of great utility in the clinic as it would help patient management. In this manuscript, Carrier and collaborators derive a signature for melanoma aggressiveness relying on methylated regions of tumour and cell line genomes. The approach the authors take is innovative as it relies on the premise that genes that make cells be more aggressive should be detected across different organisms. In their results, the authors devise a DNA methylation score that correlates with survival and can be potentially useful for patient stratification.

## Introduction

Cutaneous metastatic melanoma is the deadliest form of skin cancer, and its occurrence is growing (*Moran et al., 2018*). The recent development of targeted and immune therapies has dramatically improved patient's outcomes. Indeed, median overall survival (OS) of patients with advanced-stage melanoma has increased from ~9 months to at least 2 years since 2011 (*Luke et al., 2017*). OS is better after targeted (*Teterycz et al., 2019*) or immunotherapies (*Weiss et al., 2019*), but there are still non-responders and neo/acquired resistants. Despite these advances, there is place for improvement in particular to discover novel early prognostic markers and potential avenue for adjuvant therapies.

DNA methylation in malignant melanoma has been studied to identify specific DNA methylation changes and decipher their impact. Melanoma has a CpG island methylator phenotype (CIMP) (*Tanemura et al., 2009*), and several methylated genes are associated with melanoma progression (*Micevic et al., 2017*), with aggressive clinical and pathological features and poor survival in patients (*de Unamuno Bustos et al., 2018*; *Guo et al., 2019*), are candidate epigenetic drivers of melanoma metastasis (*Chatterjee et al., 2017*), or are implicated in immunotherapy resistance (*Emran et al., 2019*). Such DNA methylation changes have been studied at different stages of the metastatic disease but not in primary cutaneous tumour. Importantly, DNA methylation has been shown to occur very early in tumour formation (*Michalak et al., 2019*) and thus has the potential to provide early biomarkers indicating the metastatic potential of the tumour. However, the field currently lacks a genomic strategy that can both account for genetic environment and identify early DNA methylation markers that predict the aggressiveness of the melanoma.

Here, we developed a strategy that leverages the DNA methylome from different pairs of human melanoma cells lines. Cells within pairs share a common genetic background, but one counterpart has been selected for aggressiveness in different in vivo contexts (human vs murine). We proposed that the DNA methylation signature of tumour aggressiveness would be independent of the physiological context; starting from a human tumour, shared signatures relevant to aggressiveness should emerge independent on whether this trait were acquired in humans, or whether cells have been implanted into rats or mice. In a multi-step selection process, we identified hypermethylated sites common to the most aggressive melanoma forms, analysed the distribution of these sites in the genome, and validated these methylation peaks in cell lines and patient samples. This strategy identified a DNA methylation signature of five CpG sites in four gene promoters in primary tumours that could predict the OS of the patients and thus has potential diagnostic application. This strategy, which overcomes heterogeneity in tumours due to the environment, can potentially be generalized to other cancers involving DNA methylation alterations.

## Results

### A three-step strategy identifies differentially methylated genes that identify melanoma aggressiveness

To identify genes whose DNA methylation state is related to the metastatic melanoma aggressiveness, we designed a strategy to compare the DNA methylome of three pairs of melanoma cell lines. Each pair was derived from the same patient melanoma cell lines that differed in aggressiveness and microenvironmental exposure with respect to their clinical origin or subsequent in vivo experimental processing (*Figure 1*). The first pair consisted of the WM115 and WM266-4 cell lines, derived from a vertical growth phase (VGP) primary melanoma and a cutaneous metastasis from the same patient, respectively, thus comparing a less and more aggressive pair of human melanoma cells. The second and the third pairs include a cell line established from a human lymph node metastasis (M4Be) and two metastatic variants selected for their increased metastatic potential in xenograft experiments either in mouse (M4BeS2) or in rat (TW12) (*Figure 1A*, step 1). It is important to note that in each pair, the aggressive cell line is derived from the same genetic background, but the most aggressive lines emerged in different in vivo contexts: human, mouse, and rat, respectively. We hypothesized that common aggressive DNA hypermethylation signatures should emerge early in tumorigenesis and should be shared in aggressive cells, independent of the physiological context under which this trait arises. Careful cell culture practice was applied to limit cell culture process-related divergence of DNA methylation between the cell lines to be compared.

The DNA methylation profiles of each cell line was analysed using the Human Methylation 450 K array BeadChip technology to identify the hypermethylated genes in the more aggressive variants. A first global analysis showed that nearly half of the analysed genes displayed at least one CpG position, where methylation levels are increased over 20% in the aggressive cell line compared to its respective counterpart. We made the choice to restrict the analysis to CpGs in promoters or first exons to consider correlations to key biological processes associated to DNA methylation changes in cutaneous melanoma. Following this first step of selection, we adopted two complementary approaches (*Figure 1B*). An oriented strategy, based on the genomic mapping of the 229 hypermethylated genes common to the aggressive melanoma cell lines, allowed us to identify clusters of hypermethylation. Clusters consisted of at least two hypermethylated genes that are either direct neighbours or separated within 3 megabases (Mb) of one another (*Figure 2*). Sex chromosomes were excluded from this analysis because they are subject to parental imprinting (*Barlow, 2011*).

Bootstrap analysis of the repartition of the 229 genes along the chromosomes confirmed a non-random distribution, and hypermethylated genes enriched in short chromosomal regions (*Figure 2* and *Figure 2—figure supplement 1*). Nine methylation clusters were identified on chromosomes 5, 6, 10, 15, 16, and 17 (dotted line circles in *Figure 2* and list in *Figure 2—figure supplement 2*), containing a total of 74 genes (*Figure 2—figure supplement 3*). Among these genes, 34 were further selected because they displayed at least two hypermethylated CpGs located in the promoter region (TSS1500-TSS200-5'UTR-first exon) and a 40% difference in methylation when comparing human WM266-4 to WM115 cells. Chromosomes 5 and 17 are of particular interest as the methylation clusters contain large multigenic families (*Figure 2—figure supplement 1*). On chromosome 5, nine genes were identified as hypermethylated with a methylation cluster containing six genes belonging to the protocadherin beta (*PCDHB*) family (*Figure 2—figure supplement 1A* and insert). *SPAG7*, *SOCS3*, and *RAC3* displayed the strongest hypermethylation values (over 90%) among the 10 hypermethylated genes found on chromosome 17 that had at least two CpGs in the promoter region with a >40% difference methylation in WM266-4 cells. Interestingly, this methylation cluster included five members of the multigenic myosin heavy chain (MYH) family (insert in *Figure 2—figure supplement 1B*).

We then reasoned that even if not associated to gene expression, this specific high difference in DNA methylation common to different aggressive melanoma should play a role in cancer. Thus, the second strategy was non-oriented and consisted of analysing the functional pathways in which the 229 hypermethylated genes were involved. Using QIAGEN Ingenuity Pathway Analysis (IPA, QIAGEN Redwood City, https://digitalinsights.qiagen.com/) software, we found 116 genes highly associated with known functions (p<0.05, *Figure 1—figure supplement 1*). In the top 15 functions, which might play a role in aggressiveness and carcinogenesis of the melanoma cells, were cell-to-cell signalling and interactions, cellular assembly and organization, and cancer and cellular movement. Finally, when cross-checking the 34 genes from the oriented strategy and the 116 genes from the non-oriented

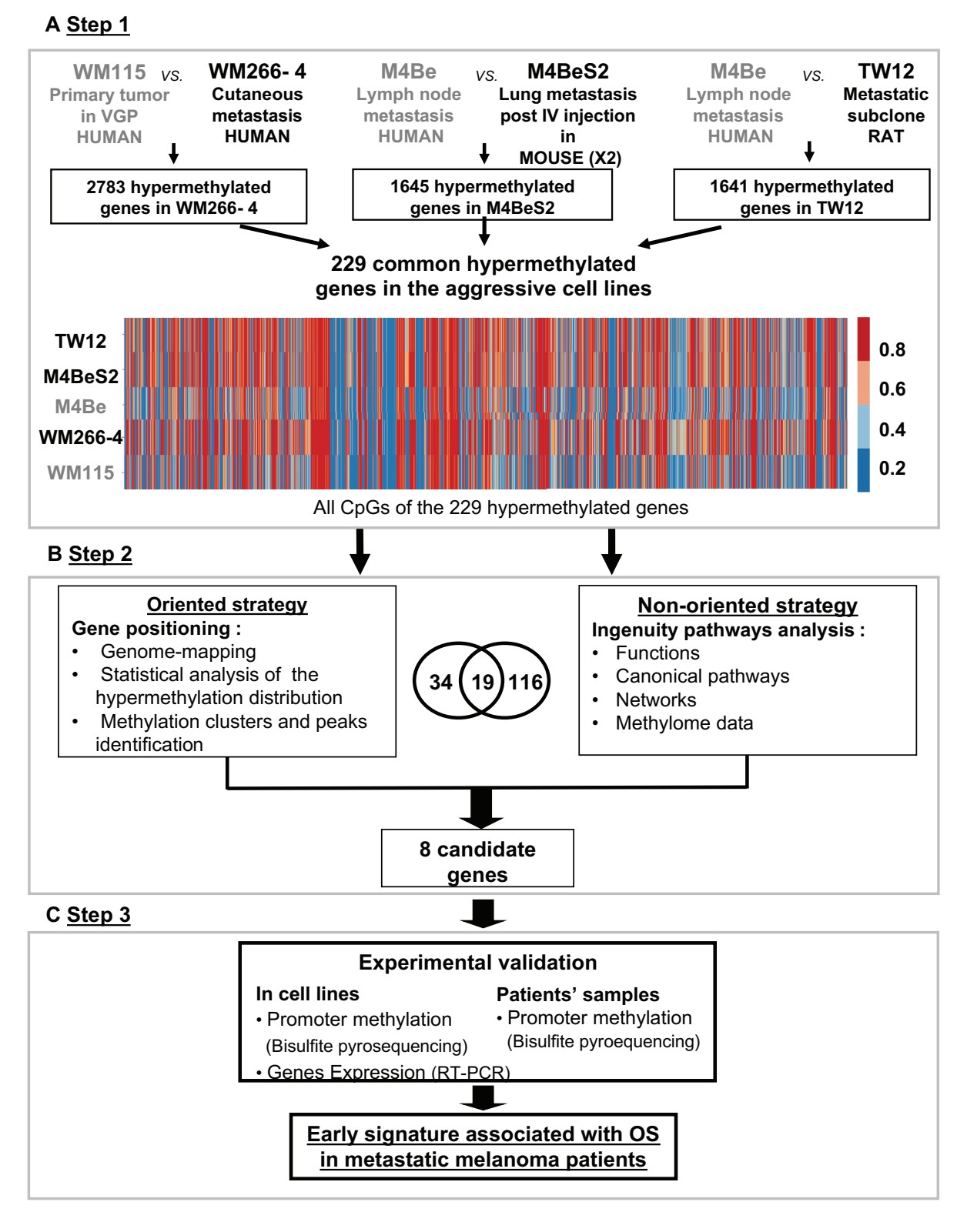

**Figure 1.** Strategy for identifying differentially methylated gene signatures of aggressive melanoma. The strategy is based on the analysis of three pairs of human melanoma cell lines with an aggressive variant derived under different physiological contexts: human, mouse, and rat. In each pair, the cell line defined as more aggressive is indicated in bold characters. (**A**) Step 1: the methylation status of more than 480,000 CpG positions was compared in each cell line pair using the Illumina Infinium Human Methylation 450 K BeadChip technology. 229 common genes showing at least three CpGs positions with

*Figure 1 continued on next page*

*Figure 1 continued*

methylation levels increased by 20% in the aggressive cell line were retained (hypermethylated genes). (**B**) Step 2: two strategies for data analysis were used: the oriented strategy is based on a statistical analysis of the distribution of the hypermethylated genes across the genome, and the non-oriented strategy uses Ingenuity Pathway Analysis software to identify potential links to described networks and functions. (**C**) Step 3: experimental validation of the selected genes, by bisulfite pyrosequencing for DNA methylation and RT-PCR for gene expression, was performed in the WM115 and WM266-4 cell lines prior to analysis in patient samples. After applying this differential threshold to at least three CpG positions for each gene, we found that 2783, 1645, and 1641 genes were hypermethylated in WM266-4 vs WM115, M4BeS2 vs M4Be, and TW12 vs M4Be, respectively (**A**). 229 genes, comprising 5590 CpG sites, were common to all three pairs of cell lines. These 229 genes were further analysed using the human WM115/WM266-4 pair. 1287 (23%) CpGs were hypermethylated (>20%) in WM266-4 cells of which 788 (61%) were located in promoter regions (TSS1500-TSS200-5'UTR-first exon), 452 (35%) in gene bodies and 47 (4%) in 3'UTR regions.

The online version of this article includes the following figure supplement(s) for figure 1:

**Figure supplement 1.** Top functions associated by Ingenuity Pathway Analysis (IPA) to the 229 hypermethylated genes in the aggressive cell lines.

**Figure supplement 2.** Detailed information of the eight selected genes.

strategy, we identified 19 common genes (*Figure 1B*). Notably, all of 19 genes were associated with one or two of the top functional networks from the IPA (*Figure 1—figure supplement 1* and *Supplementary file 1*).

## Gene selection and validation

The third part of our approach consisted of validating the methylation status of a subset of these candidate genes in melanoma cell lines and patient's tissues samples. Combining the cluster analysis and the IPA results, 19 overlapped and we chose in total eight genes because: (*Moran et al., 2018*) distributed on four different chromosomes, bearing hypermethylation clusters (*Luke et al., 2017*), represent hypermethylation peaks, showing a strong methylation difference between the aggressive and non-aggressive cell lines, and *Teterycz et al., 2019* have a potential role in aggressiveness suggested by the literature (*Figure 1—figure supplement 2*). On chromosome 17, we chose the following four genes. *MYH1* was selected because it forms a highly differentially methylated cluster (*Figure 2—figure supplement 1B*) and is known to show aberrant expression levels in aggressive cells in head and neck squamous and lung carcinoma tumours (*Vachani et al., 2007*). *SOCS3* and *RAC3* displayed among the highest hypermethylation peaks in our aggressive melanoma lines (92 and 96%, respectively), with a very high differential methylation score, above 70%, between WM266-4 and WM115 cells. In addition, *SOCS3* has previously been reported to be hypermethylated in melanoma (*Tokita et al., 2007*), while loss of *RAC3* expression has been associated with impaired invasion in glioma and breast carcinoma cells (*Chan et al., 2005*; *Baugher et al., 2005*). *HOXB2* was chosen for its lower methylation score (69%) and differential methylation score (45%). It has previously been associated with progression of bladder cancer when silenced by promoter hypermethylation, and it can be re-expressed upon demethylation treatment with 5-azacitidine (5AzadC) (*Marsit et al., 2010*). On chromosome 5, two genes were chosen from the PCDHB hypermethylation cluster: *PCDHB15* and *PCDHB16* that are CIMP-associated with bad prognosis in neuroblastoma (*Abe et al., 2005*; *Banelli et al., 2012*). Of note, neurons and melanocytes originate from the same germ layer during embryogenesis. Furthermore, *BCL2L10* (B cell lymphoma 2 like 10) located on chromosome 15 was selected because it bears the highest methylation score (73%) on this chromosome, it is implicated in apoptosis, it was previously described as hypermethylated in gastric cancer cell lines (*Mikata et al., 2010*), and associated with poor prognosis in gastric cancer patients (*Xu et al., 2011*; *Voso et al., 2011*). Finally, *MIR155HG*, located on chromosome 21, encodes a microRNA, miR-155, was chosen because linked to cell proliferation and cancer (*Elton et al., 2013*), including in melanoma where it is downregulated (*Levati et al., 2009*; *Wang et al., 2020*).

The methylation status of these eight genes (*MYH1*, *RAC3*, *SOCS3*, *HOXB2*, *PCDHB15*, *PCDHB16*, *BCL2L10*, and *MIR155HG*) was further validated in WM115 vs WM266-4 cell lines by DNA pyrosequencing after bisulfite conversion and PCR amplification on 100 bp regions containing the CpGs identified in step 2 (*Figure 1C*). Seven genes pass the threshold of validation, 20% DNA methylation difference between the cell lines (*Figure 3—figure supplement 1*).

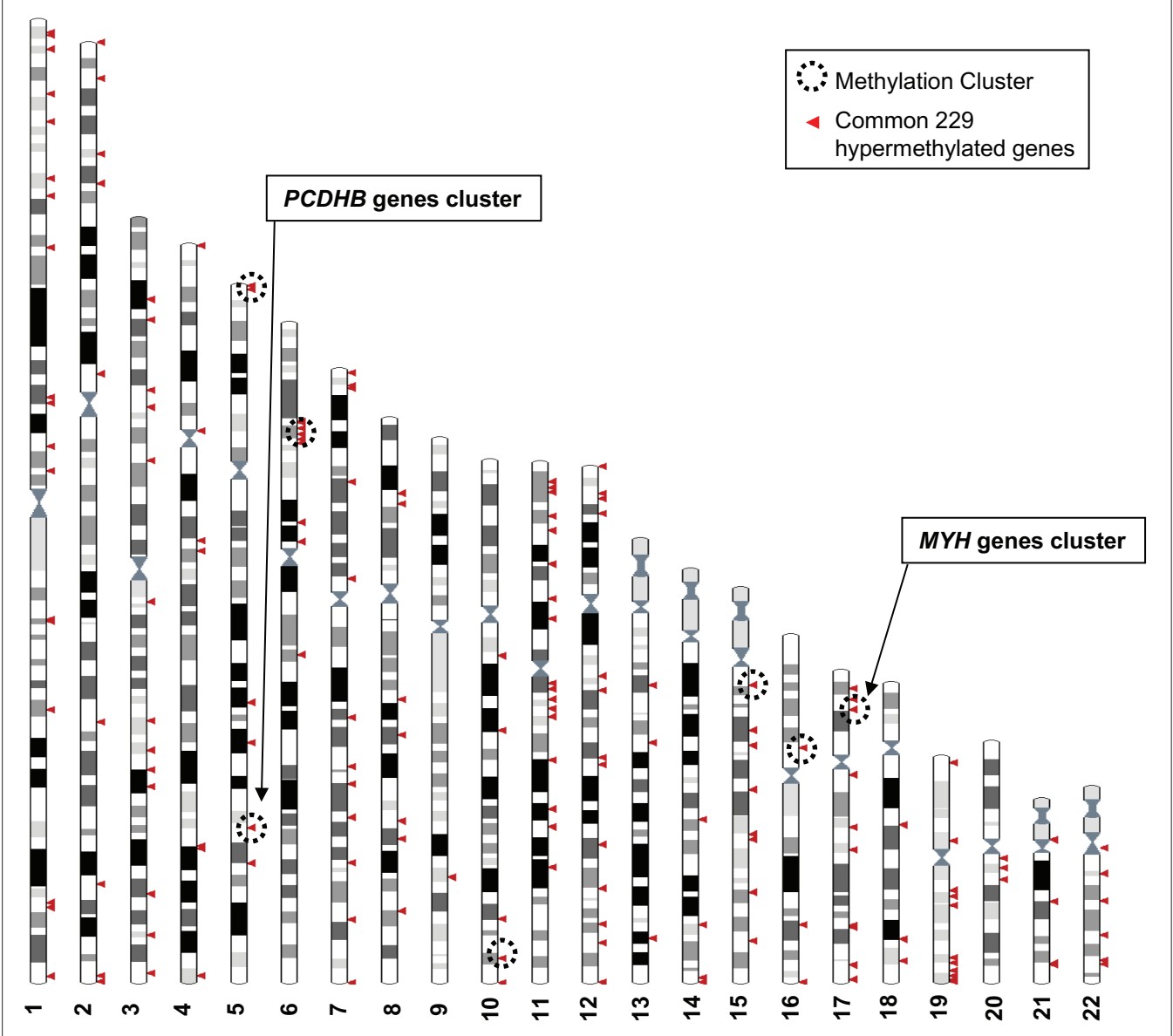

**Figure 2.** Genomic distribution of the 229 commonly hypermethylated genes in the more aggressive cell lines. The Ensembl genome browser (http://www.ensembl.org, view on karyotype) was used to map the 229 hypermethylated genes to the human genome. Sex chromosomes were excluded from the analysis. Each arrowhead could correspond to several genes. Methylation clusters are indicated by dotted line circles. Chromosome 5 and 6 circles correspond to two clusters that are too close to be separated on this scale.

The online version of this article includes the following figure supplement(s) for figure 2:

**Figure supplement 1.** Examples of genomic localization of the genes hypermethylated in the most aggressive cell lines.

**Figure supplement 2.** Clusters of hypermethylated genes identified by the oriented strategy.

**Figure supplement 3.** List of the 74 hypermethylated genes found on six chromosomes bearing at least one cluster.

## Validation in patient samples and identification of a methylation signature

Next, we assessed the methylation profile of the eight selected genes in 20 tumour tissues from melanoma patients of which 10 were from metastatic melanomas and 10 were from primary melanomas (*Figure 1C*). Four genes (*MYH1*, *PCDHB16*, *PCDHB15*, and *BCL2L10*) showed a differential methylation profile between metastatic and primary tumour tissue samples (data not shown). For further validation, CpG sites in these four genes were individually analysed using bisulfite conversion followed

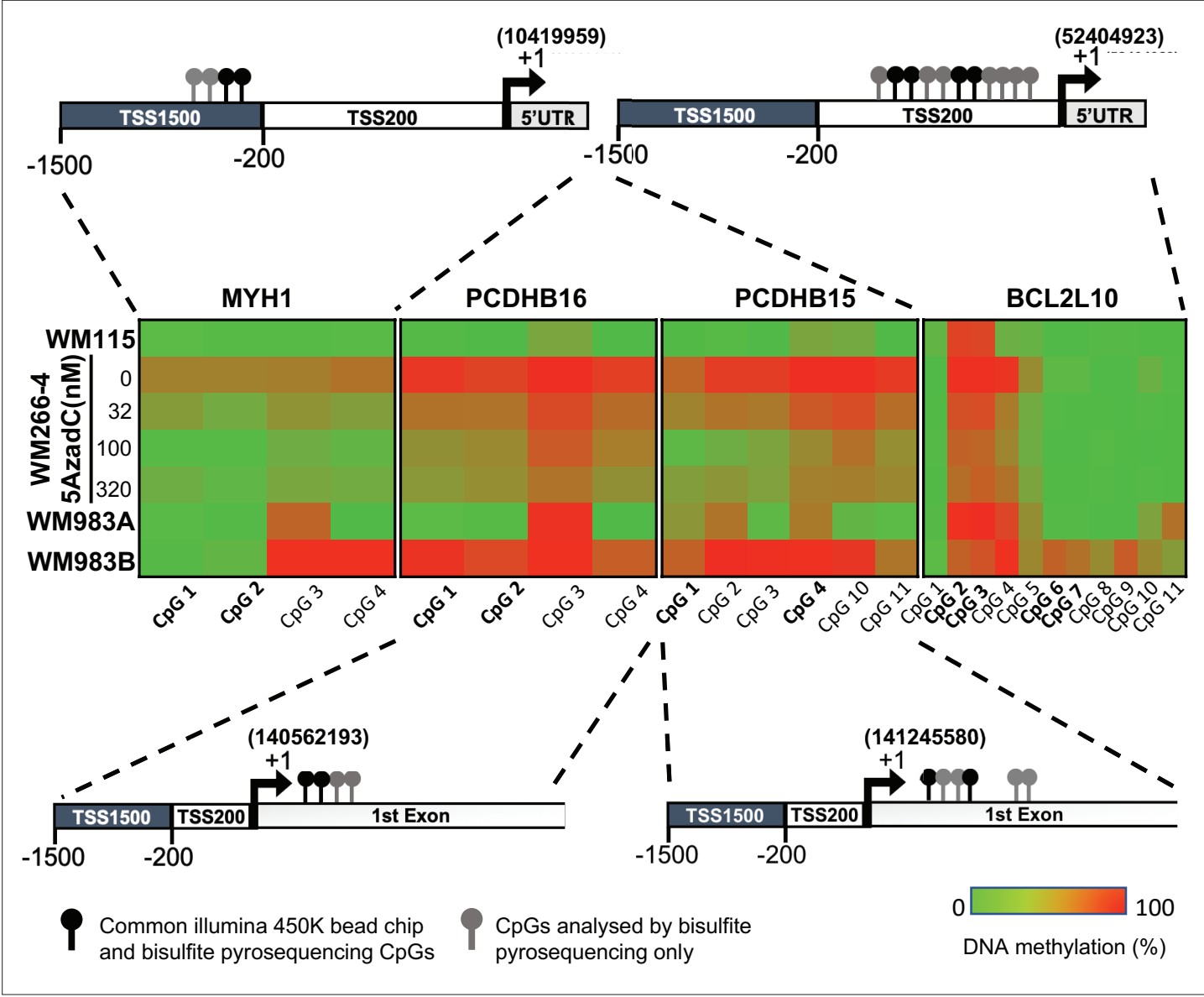

**Figure 3.** Promoter hypermethylation of the candidate genes in cell lines. (**A**) Shown is the localization of all the CpGs analysed (lollipops) on the four selected genes (*MYH1*, *PCDHB15*, *PCDHB16*, and *BCL2L10*). The CpGs present on the 450 k array are in bold type (black lollipops). The heatmap indicates the DNA methylation percentage (red = 100 and green = 0) of the indicated CpGs in each gene and each cell line. WM115 and WM983A are primary cell lines derived from two patients. WM266-4 and WM983B are the cutaneous and lymph node metastasis counterpart. WM266-4 cells were treated with daily doses of 5AzadC (32, 100, and 320 nM) 72 hr before genomic DNA extraction.

The online version of this article includes the following figure supplement(s) for figure 3:

**Figure supplement 1.** Promoter hypermethylation of the candidate genes in WM266-4 vs WM115 cells.

by pyrosequencing in reference pair of cell lines (WM115/WM266-4) as well as two melanoma cell lines derived from the same patient: WM983A (primary site) and WM983B (lymph node metastatic site, *Figure 3*).

DNA methylation levels at each CpG site were analysed by bisulfite pyrosequencing, confirming a clear difference in methylation for the four genes in the aggressive tumour cells compared to less aggressive forms. Interestingly, this DNA methylation could be reversed in WM266-4 cells using a low dose of 5AzadC treatment (32, 100, and 320 nM). The median methylation of these individual CpGs was determined on the first set of patient samples, 10 metastatic and 10 primary tumours, and on additional 10 primary tumour samples (*Figure 4—figure supplement 1*). Remarkably, the median of

DNA methylation within primary samples appeared to be inversely correlated with patient OS. Thus, we defined patients with primary tumours diagnosis and long survival (LS) with an OS >1 year and patients with short survival (SS) with an OS (median survival = 6 months) ≤1 year (median survival = 51 months) after diagnosed. 1 year was chosen because, at the time of the sampling and the beginning of the study, it was the average OS of diagnosticated cutaneous melanoma. In addition, the delay of 1 year well distinguished the two group of LS and SS. The DNA methylation profile in primary tumours with SS similar to that observed in metastatic patients (top, in red, *Figure 4—figure supplement 1*). The analysis showed that *MYH1* was globally hypomethylated in SS patients, whereas *PCDHB16*, *PCDHB15*, and *BCL2L10* were hypermethylated. Data from 29 additional primary tumours samples were then analysed. The extended cohort of patient with primary melanoma primary (n = 49) confirmed that these four differentially methylated genes represent specific markers of more aggressive SS primary and metastatic melanoma tumours (*Figure 4A*).

To determine the combination of CpGs that might predict melanoma aggressiveness and thus survival outcome in patients with primary tumours, DNA methylation at individual CpG sites was analysed using bisulfite pyrosequencing (CpG positions are illustrated in *Figure 3* and detailed in *Supplementary file 2*). One CpG showed significantly differential methylated in primary samples from SS patients (median SS = 6 months) when compared to LS patients (median LS = 51 months) for *PCDHB15*, *PCDBH16*, and *MYH1*, and two CpGs for *BCL2L10* (*Figure 4B*). DNA methylation *MYH1* CpG was inversely correlated with survival duration and was significantly hypomethylated in primary samples from SS patients; whereas *PCDHB16*, *PCDHB15*, and *BCL2L10* CpGs were hypermethylated in SS patients. To validate the robustness of this signature, a score was calculated as follows: a score of 1 was given to each gene when the median DNA methylation of the CpG met conditions for hypomethylation (>15%, seen for *MYH1*) and hypermethylation (>15%, for *PCDHB16*, *PCDHB15*, and *BCL2L10*). CpGs not meeting these criteria were given a score of 0. The mean of the two CpGs was considered for BCL2L10 as they are close to each other and correspond to a methylation peak. The final score was obtained by summing points attributed to each gene so that scores range from 0 to 4 (*Supplementary file 3*). A threshold of 2 (i.e. with at least two genes matching this condition) was used to include patients with a methylation score ≥2. The methylation score was represented as a function of survival in months (*Figure 4C* and *Figure 4—figure supplement 2A*). Patients with a methylation score ≥2 (red line) had a shorter life expectancy (≤1 year) than those with a methylation score value <2 (blue line). This analysis demonstrated that a methylation score of at least 2 in primary melanoma samples is predictive of patient outcome (log-rank test, p=0.0008) with a significant hazard ratio of 3.4 (p=0.001, concordance index = 0.62; *Figure 4—figure supplement 2B*). Then, we compared the methylation score to the clinical parameter used in clinic, the Breslow index. Based on the melanoma American Joint Committee on Cancer (AJCC) staging (T1, T2, T3, and T4), we confirmed that on our 49 patient samples, an increased Breslow depth of the primary tumour is a prognostic factor for survival probability (Kaplan–Meier plot in *Figure 4—figure supplement 2C*). As our cohort contained only one sample of T1 grade (primary tumour's depth less than 1 mm), we then compared the survival probability between two groups: Breslow index below 2 mm (T1 and T2) or above 2 mm (T3 and T4). We obtained no statistical difference between the two groups (*Figure 4—figure supplement 2D* and E) in contrast to what obtain upon use of the methylation signature (*Figure 4—figure supplement 2A and B*). Most interestingly, the interaction between the methylation score and the Breslow index gave a significant increase of the hazard ratio from 3.4 to 6.3 (p<0,001, concordance index = 0.63, *Figure 4—figure supplement 2F*). This result support that the methylation score could improve the clinical prognostic utility of the Breslow index.

## Discussion

To the best of our knowledge, the multistep strategy that we developed and used to identify differentially methylated genes in melanoma cells predicting aggressiveness is original. It is important to underline the starting point. We reasoned that since epigenetic changes are linked to cell plasticity and biological environment changes (*Hanahan and Weinberg, 2011*; *Feinberg and Fallin, 2015*; *Feinberg et al., 2006*), any common DNA methylation changes acquired by the most aggressive cells in different in vivo contexts (human, mouse, and rat) would highlight a robust and relevant trait of melanoma tumour cell plasticity and aggressiveness despite their heterogeneity. Indeed, melanoma is highly heterogeneous tumour, and until now it has been difficult to find a signature in primary

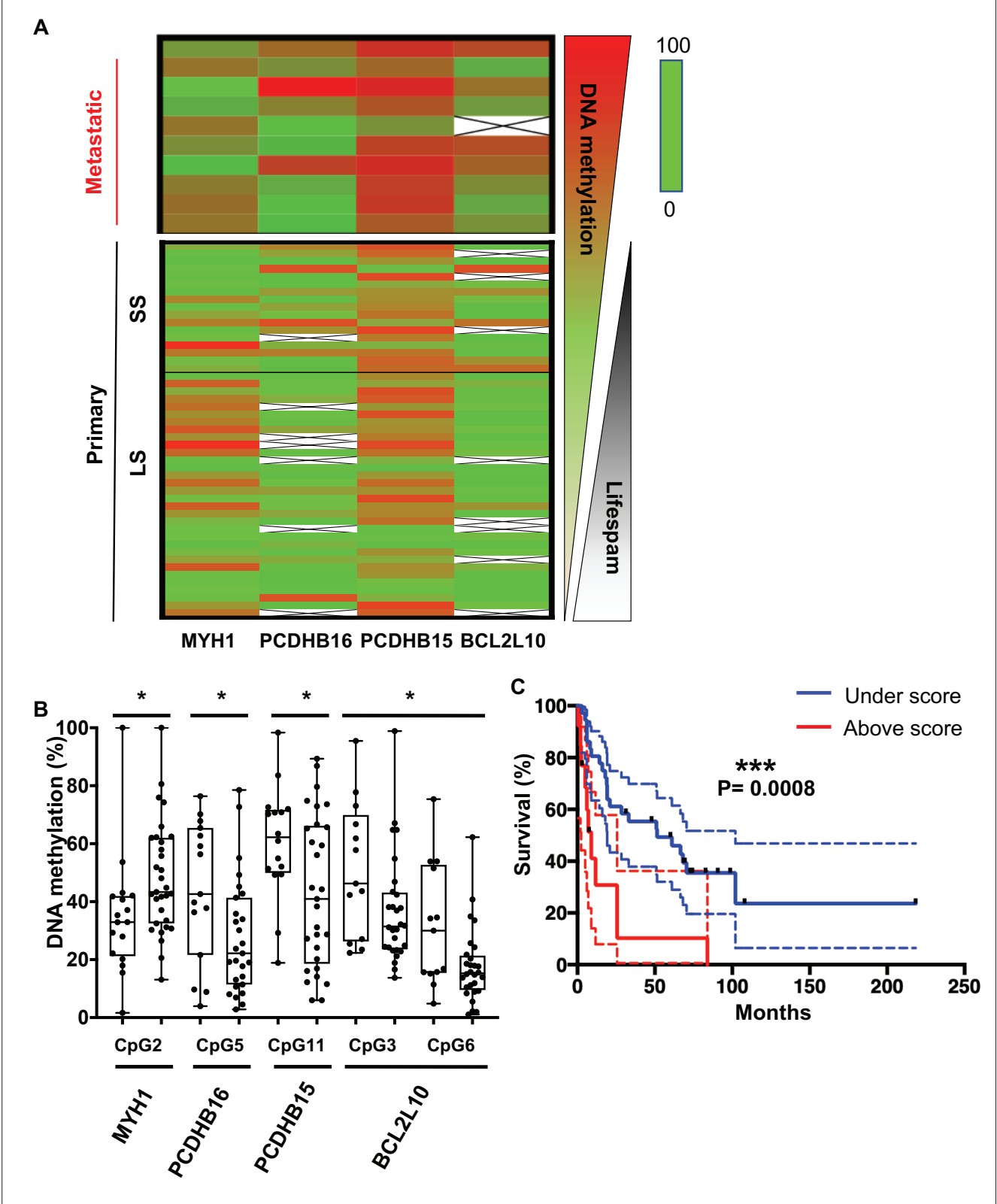

**Figure 4.** CpGs DNA methylation in primary melanomas predicts patient outcome. (**A**) The heatmap shows the median DNA methylation of analysed CpGs for each gene in metastatic (n = 10) and primary (n = 49) patient samples. Primary samples are divided by a black line indicating the cut-off at 1 year between the short survival and long survival (SS [left] and LS [right]). (**B**) DNA methylation changes in selected CpGs between SS (left box) and LS (right box) patients, respectively (n = 49). Fisher test to analyse variances and t-test were performed, *p<0.05. (**C**) Kaplan–Meier curve of patients'

*Figure 4 continued on next page*

*Figure 4 continued*

survival based on the methylation score calculated for the five CpGs reported in panel B. The methylation score = 2 corresponds to at least two CpGs that showed a methylation difference >15% when compared to the DNA methylation median of the primary metastasis.

The online version of this article includes the following figure supplement(s) for figure 4:

**Figure supplement 1.** Heatmap of DNA methylation of MYH1, PCDHB16, PCDHB15, and BCL2L10 in the first set of 10 metastatic melanoma patient sample and 20 primary patient melanoma.

**Figure supplement 2.** The DNA methylation score is a better prognostic factor than classical used clinical parameter Breslow index.

melanoma, either based on DNA methylation or gene expression that help to predict prognosis as well as Breslow index (*Bhalla et al., 2019*). Several DNA methylation studies have been conducted comparing patient samples at different stages of cutaneous melanoma to normal samples, i.e., melanocytes or nevi (*Marzese et al., 2014*; *Gao et al., 2013*; *Sigalotti et al., 2012*; *Marzese et al., 2015*) or using cell lines derived from multi-grade patient samples (*Li et al., 2013*; *Chatterjee et al., 2017*). While such studies have identified genes regulated by DNA methylation, none have yet identified a common pattern or a specific signature of melanoma aggressiveness. In contrast, the unique approach used in our study yielded a potential DNA methylation signature that correlates with outcomes. We report the following main and original results: (i) the observation that robust DNA methylation traits of aggressiveness are independent of the physiological context; (ii) the methodology of combining DNA methylome analysis and chromosome cluster-based analysis that can be applied beyond melanoma; and (iii) the identification of the methylation of five CpGs (and not genes) that provide a predictive value of the aggressiveness of the melanoma in primary tumours.

Considering that the cellular models utilized have experienced very different experimental microenvironments, we assumed that the common 229 identified hypermethylated genes represented genes either playing a direct role in or being associated with the aggressiveness of melanoma. Next, we applied a bootstrap analysis of the distribution of these hypermethylated genes revealing that they are not randomly distributed along the chromosomes but are instead organized in clusters of hypermethylated genes. This observation of clustered genes regulated by DNA methylation is reminiscent of the mechanism underlying parental imprinting and its spreading on imprinted genes during development (*Lewis and Reik, 2006*; *Lindsay and Adams, 1996*). It is also coherent with the association of hypermethylated regions with long range epigenetic silencing (LRES) described by *Frigola et al., 2006* in colon cancer and with the fact that LRES regions containing hypermethylated genes extend up to 4 Mb and are correlated with gene extinction (*Coolen et al., 2010*). Thus, guided by the hypothesis that these clustered genes correspond to early changes in CpG methylation during tumour progression and may thus constitute a starting point for spreading of DNA methylation, we concentrated our efforts on these. We concentrated our efforts on CpGs that could play a role in cancer progression, and an IPA of the corresponding biological functions related to tumorigenesis or resistance to therapy facilitated selection of eight potential genes that could be early signals of the aggressiveness of the disease. The DNA methylation signature at these eight genes was analysed in additional melanoma cell lines (WM983A and WM983B) derived from the same patient and in patient metastatic and primary tumour samples, highlighting four genes of interest. All four genes are demethylated upon treatment with 5AzadC in WM266-4 metastatic melanoma cells and displayed the strongest methylation differential.

In the last part of our study, we challenged the methylation status of these four gene promoters in 59 patient samples. Surprisingly, five CpGs were significantly differentially methylated between patients with a short OS (<1 year, SS) and those with a longer OS (>1 year, LS). This led to the discovery of an early and robust signature of melanoma progression that is only based directly on the primary tumour that can significantly predict patient outcome (p=0.0008). Several studies have described different methylation patterns at various stages of melanoma disease, but only one reported a difference in methylation profile among primary tumour samples linking it to an ulceration status and thus a poor clinical outcome (*Rakosy et al., 2013*). More recently, *Guo et al., 2019* identified four DNA methylation biomarkers by analysing all melanoma types in the The Cancer Genome Atlas (TCGA) database but without validating this on another patient cohort. Importantly, we have discovered five CpG sites that are key to metastatic melanoma formation and are grouped in genomic clusters. This signature is unique, as is the integrated approach and the baseline assumption used to identify it. Importantly,

patients samples were from different regions and continents (France, Italy, Portugal, and USA). Going forward, the signature could be further developed to predict the much-needed progression risk of the melanoma directly in primary tumours. For example, the methylation level of the five CpGs combined with the Breslow score (*Figure 4—figure supplement 2*) could be explored to confirm if it can be applied to identify the patients that are at risk when primary tumours are collected and removed.

Recently, a DNA methylation signature based on four CpGs was shown to prognostic biomarker for osteosarcoma (*Lyskjær et al., 2022*). Of note, diagnostic kits based on DNA methylation exist for colon, lung, and prostate cancer (*Payne, 2010*; *Kneip et al., 2011*; *Li et al., 2014*).

In parallel, the DNA methylation analysis allowed us to identify that miR199-3p is silenced by promoter hypermethylation promoting metastasis formation (*Desjobert et al., 2019*) and that PCDHB15 plays a role in inhibiting invasion, aggregation, and lung metastasis formation in vivo when stably overexpressed in metastatic melanoma cells (study ongoing).

## Conclusions

We developed a novel multistep approach that allowed us to identify a methylation signature of five CpGs in primary melanoma tissues that has the potential to predict survival outcomes in cutaneous melanoma patients. Importantly, the five CpGs are located on four different genes, and it is not the expression of the genes that is monitored. Our method was based on two main concepts; the first one being that aggressive traits marked by DNA hypermethylation appear early in the disease and are independent of physiological context. The second concept is that hypermethylated sites in metastatic forms of melanoma are gathered in genomic clusters. On the methodological side, we combined analysis of the DNA methylome with chromosomal location. Following these general concepts, this integrated approach can be applied not only to other cancer types but also to other diseases or biological processes where DNA methylation changes are identified.

# Materials and methods

**Key resources table**

| Reagent type (species) or resource | Designation | Source or reference | Identifiers | Additional information |
|---|---|---|---|---|
| Cell line (*Homo sapiens*) | WM115 | ATCC | CRL-1676 | |
| Cell line (*Homo sapiens*) | WM-266–4 | ATCC | CRL-1675 | |
| Cell line (*Homo sapiens*) | WM983A | Coriell Institute | WC00048 | |
| Cell line (*Homo sapiens*) | WM983B | Coriell Institute | WC00066 | |
| Cell line (*Homo sapiens*) | M4Be | *Bailly and Doré, 1991* | Not applicable | |
| Cell line (*Homo sapiens*) | TW12 | *Bailly and Doré, 1991*; *Bertucci et al., 2007*; *Thomas et al., 1995* | Not applicable | |
| Cell line (*Homo sapiens*) | M4BeS2 | *Clark et al., 2000* | Not applicable | |
| Biological sample (*Homo sapiens*) | Department of Pathology, IUCT-O Toulouse Hospital (France) | | Primary melanomas (n=12), lymph node metastases (n = 7), and cutaneous metastases (n=3) | |
| Biological sample (*Homo sapiens*) | Department of Experimental Oncology, European Institute of Oncology, Milan (Italy) | | Primary melanoma (n = 5) | |
| Biological sample (*Homo sapiens*) | Saint John's Cancer Institute (formerly John Wayne Cancer institute [USA]) | | Primary melanomas (n=12) | |
| Recombinant DNA reagent | Department of Pathology of the Portuguese Oncology Institute of Porto (IPO-Porto) | | Primary melanomas (n=20) | |
| Commercial assay or kit | DNeasy Tissue kit | Qiagen | 69504 | |
| Commercial assay or kit | QiaAmp kit | Qiagen | 965672 | |

*Continued on next page*

*Continued*

| Reagent type (species) or resource | Designation | Source or reference | Identifiers | Additional information |
|---|---|---|---|---|
| Commercial assay or kit | FFPE RNA/DNA Purification Plus Kit | FFPE RNA/DNA Purification Plus Kit | 54300 | |
| Commercial assay or kit | EpiTect 96 Bisulfite Kit | Qiagen | 59104 | |
| Commercial assay or kit | Infinium Human Methylation 450 K BeadChips | Illumina | WG-310–1003 | |
| Commercial assay or kit | PyroGold SQA reagent kit | Qiagen | 972824 | |
| Recombinant DNA reagent | HotStarTaq DNA polymerase | Qiagen | 203207 | |
| Software and algorithm | GenomeStudio | Illumina | Version 2011.1 | |
| Software and algorithm | R studio | RStudio, Inc | RStudio 2021.09.2 382 'Ghost Orchid' Release. R version 4.1.2 | |
| Software and algorithm | PyroMark software | Qiagen | V1.0 | |
| Software and algorithm | GraphPad Prims8 | Dotmatics | Version 8 | |
| Software and algorithm | QIAGEN's Ingenuity Pathway Analysis | Qiagen | 836508 | |

## Cell culture

The WM115 and WM266-4 cell lines were obtained from the American Type Culture Collection. The WM-115 cell line was derived from a human primary melanoma in early stages of the VGP. The WM266-4 cell line was derived from a cutaneous metastatic melanoma tumour of the same patient. These two cell lines were derived from the same female patient, with a similar sampling age (55 years old). The WM983A and WM983B cell lines were obtained from the Coriell Institute (USA). The WM983A was derived from a human primary melanoma in VGP, and the WM983B is the cutaneous lymph node metastasis in same male patient (both sampled at 54 years old). The M4Be cell line was established from a human cutaneous lymph node metastasis (*Jacubovich and Doré, 1979*). The TW12 cell line is a human aggressive variant of the parental M4Be cell line that was obtained in vivo after two serial transplantations (subcutaneous xenografts in new-born immuno-deprived rats). A subclone (TW12) was selected after limiting dilution for its high ability to form lung metastasis (*Bailly and Doré, 1991*; *Bertucci et al., 2007*; *Thomas et al., 1995*). The M4BeS2 cell line was obtained in the L Lamant's laboratory according to the in vivo selection scheme described by *Clark et al., 2000*. Briefly, M4Be cells were xenografted intravenously in nude mice. Lung metastases were collected, grown briefly in vitro, and used for a second cycle of intravenous injection. Lung metastases were collected and established in vitro as the M4BeS2 human cell line. WM983A and WM983B cell lines were grown in 20% Leibovitz L-15 medium (v/v), 2% foetal bovine serum (FBS) heat inactivated (v/v), 5 µg/mL insulin, and 1.68 mM CaCl$_2$. All other cell lines were grown in Dulbecco's Modified Eagle Medium (DMEM) (Invitrogen, France) supplemented with 10% FBS (Sigma, France), 2 mM glutamine, 100 IU/mL penicillin-streptomycin, and 1.25 µg/mL fungizone (Invitrogen) in 5% CO$_2$. Quantitation of viable cells was performed using an automated Cell Viability Analyzer (Beckman Coulter Vi-Cell). All cell lines were stored in ampoules in liquid nitrogen after receipt. All cell lines were regularly verified for mycoplasma contamination using the MycoAlert Mycoplasma Detection Kit (Lonza, Switzerland) and kept for a limited number of passages in culture. Careful cell culture practice and experimental planning were applied to limit the number of passages and cell culture process-related divergence of DNA methylation between the cell lines to be compared.

## Tumour samples

Tumour samples from melanoma patients were obtained from the tumour tissue bank at the Department of Pathology, IUCT-O Toulouse Hospital (France). The study was carried out in accordance with the institutional review board-approved protocols (CRB, AC-2013–1955), and the procedures followed were in accordance with the Helsinki Declaration. Pathological specimens consisted of primary melanomas (n=12), lymph node metastases (n=7), and cutaneous metastases (n=3). Only four patients have primary and metastasis associate; all others are independents. Additional primary melanoma

**Table 1.** Clinical pathological features of primary melanoma patients.

| Variables | n (%) |
|---|---|
| Mean age (SD) | 66.40 (15.88) |
| **Gender** | |
| Male | 26 (53) |
| Female | 23 (47) |
| **American Joint Committee on Cancer eighth stages** | |
| IV | 12 (24) |
| III | 12 (24) |
| IIIA | 1 (2) |
| IIIB | 2 (4) |
| IIIC | 2 (4) |
| II | 2 (4) |
| IIA | 3 (6) |
| IIB | 6 (12) |
| IIC | 6 (12) |
| IB | 2 (4) |
| unknown | 1 (2) |
| **Mutations** | |
| *NRAS* | 1 (2) |
| *BRAF* | 6 (12) |
| Unknown | 42 (86) |

(n = 5) frozen samples were provided by the Department of Experimental Oncology, European Institute of Oncology, Milan (Italy). The Saint John's Cancer Institute (formerly John Wayne Cancer institute [USA]) Formalin-fixed paraffin-embedded (FFPE) specimen cohort included tissues including primary melanomas (n = 12). A total of 20 FFPE tissues samples from patients diagnosed with cutaneous melanoma between 2007 and 2017 at the Portuguese Oncology Institute of Porto (IPO-Porto) without any neoadjuvant treatment was included in this study. All samples were archived at the Department of Pathology of IPO-Porto. All cases were reviewed by an experienced pathologist and staged according to the eighth edition AJCC system (*Gershenwald et al., 2017*). Relevant clinical data was collected from medical charts. For DNA extraction, a 4 µm section was cut from a representative tissue block and stained with hematoxylin-eosin. Tumour areas containing >70% transformed cells were delimited, enabling macrodissection in eight consecutive 8 µm sections. This study was approved by the institutional ethics committee of IPO Porto (CES-IPOP-FG13/2016). Anonymized clinical information for all the melanoma patients analysed is available, and the clinical pathological features of primary melanoma patients are indicated below (*Table 1*).

## Genomic DNA isolation

Genomic DNA from cell lines was performed using the DNeasy Tissue kit (Qiagen, France). Genomic DNA from frozen patient samples was isolated using the QiaAmp kit (Qiagen, France). DNA extraction from FFPE sections was performed using the FFPE RNA/DNA Purification Plus Kit (Norgen Biotek, Thorold, Canada) in accordance with manufacturer's instructions. DNA concentration and purity were determined using the NanoDrop Lite spectrophotometer (NanoDrop Technologies, Wilmington, DE, USA).

## Illumina methylation 450K microarray analysis

Genome-wide DNA methylation analysis was performed on three independent samples from each cell line. 1 µg of DNA was bisulfite-treated using the EpiTect 96 Bisulfite Kit (Qiagen GmbH, Germany). 200 ng of bisulfite-treated DNA was analysed using Infinium Human Methylation 450 K BeadChips (Illumina Inc, CA, USA). The array allows the interrogation of more than 485,000 methylation CpG sites per sample covering 99% of RefSeq genes, with an average of 17 CpG sites per gene region distributed across the promoter, 5'-UTR, first exon, gene body, and 3'-UTR.

The samples were processed according to the manufacturer's protocol at the genotyping facility of the Centre National de Génotypage (Evry, France) without any modification to the protocol. We used the GenomeStudio software (version 2011.1; Illumina Inc) for the extraction of DNA methylation signals from scanned arrays (methylation module version 1.9.0, Illumina Inc). Methylation data were extracted as raw signals with no background subtraction or data normalization. The obtained 'ß' values, i.e., the methylation scores for each CpG range from 0 (unmethylated, U) to 1 (fully methylated, M) on a continuous scale, were calculated from the intensity of the M and U alleles as the ratio of fluorescent signals (ß = [Max(M,0)]/[Max(M,0)+Max(U,0)+100]).

All pre-processing, correction, and normalization steps were performed using an improved version of the in-house developed pipeline using subset quantile normalization based on the relation to sequence annotation provided by Illumina (*Touleimat and Tost, 2012*). Probes were considered as differentially methylated if the absolute value of the difference between robust median ß-values in samples of each phenotypes was higher than 0.2: median cell line 1 ($\beta_1$, $\beta_2$ and, $\beta_3$) – median cell line 2 ($\beta_1$, $\beta_2$ and, $\beta_3$) $\geq$0.2, where $\beta_1$, $\beta_2$, and $\beta_3$ correspond to the ß-values in three replicates within each cell line, all with a detection p-value<0.01. This 0.2 threshold, representing approximately a difference in DNA methylation levels of 20%, corresponds to the recommended differences between samples analysed with the Illumina methylation Infinium technology that can be detected with a 99% confidence.

Differential DNA methylation markers were identified using a combination of two approaches. The performance of individual CpGs was assessed testing the absolute DNA methylation difference between samples of the two phenotypes of interest with different thresholds and permitting a small number of misclassifications. At the same time, a vector quantization method (nearest centroid classifier) was used to define CpGs that separate, at a given threshold, the two phenotypes of interest. CpGs that were significant in both tests were used to calculate a vector using a directed z-score, which was subsequently used to assign new samples to their phenotypic group.

The corresponding genes were obtained from a list of differentially methylated probes using the Illumina annotation file, and overlap between gene lists from the three cellular pairs was determined.

The promoter methylation scores (%) reported in *Figure 2—figure supplement 1* were defined as follows: mean (probe 1 ß-value to probe n ß-value) ×100, where probe 1 to probe n are probes that are differentially methylated between WM266 and WM115 cells (with a difference threshold 0.2 on a scale from 0 to 1) and located in the promoter region (TSS1500-TSS200-5′UTR-first exon).

## Bisulfite pyrosequencing

Quantitative DNA methylation analysis was performed by pyrosequencing of bisulfite-treated DNA as described in *Tost and Gut, 2007*. CpGs for validation were amplified using 20 ng of bisulfite-treated human genomic DNA and 5–7.5 pmol of forward and reverse primer, one of them being biotinylated. Oligonucleotide sequences for PCR amplification and pyrosequencing are given in the supplementary data (*Supplementary file 2*). Reaction conditions were 1×HotStar Taq buffer (Qiagen) supplemented with 1.6 mM MgCl$_2$, 100 µM dNTPs, and 2.0 U HotStar Taq polymerase (Qiagen) in a 25 µL volume. The PCR program consisted of a denaturing step of 15 min at 95°C, followed by 50 cycles of 30 s at 95°C, 30 s at the respective annealing temperature and 20 s at 72°C, with a final extension of 5 min at 72°C. A total of 10 µL of PCR product was rendered single-stranded as previously described, and 4 pmol of the respective sequencing primers were used for analysis. Quantitative DNA methylation analysis was carried out on a PSQ 96MD system with the PyroGold SQA Reagent Kit (Qiagen) and results were analysed using the PyroMark software (V.1.0, Qiagen).

Percentages of methylation (% CpG) were measured for each individual CpG present in the regions analysed by pyrosequencing. The regions chosen were around the CpGs identified by the Illumina methylation 450 K analysis and include other CpGs. DNA methylation heatmaps were obtained using Prism8 software. The heatmaps in *Figure 3* refer to the median of the median of the methylation percentages of the n CpG analysed by pyrosequencing (median [CpG1:%,…CpGn:%]). For each analysed gene, the difference of methylation percentages of gene promoter regions comparing WM266-4 and WM155, reported in *Figure 3—figure supplement 1A*, was calculated as follows: median ([CpG1:%,…CpGn:%] in WM266-4 cells) – median ([CpG1:%,…CpGn:%] in WM115 cells).

## Definition of the signature score

The signature score considered the individual methylation values (percentages) of the selected single CpGs associated with *MYH1*, *PCDHB16*, *PCDHB15*, and the mean methylation values for the two CpGs selected for *BCL2L10*. For each gene, these methylation values were compared to the methylation median calculated from all the primary samples. A score of 1 was attributed to the gene when the methylation values differed by at least 15%. For *MYH1*, for which hypomethylation was associated with aggressiveness, this score was attributed when the methylation value was inferior to the median. Conversely, a score of 1 indicated a methylation value superior to the median for the three other genes (*PCDHB16*, *PCDHB15*, and *BCL2L10*). The signature score was the sum of the scores attributed individually to the four genes and fell between 0 and 4. Signature score and survival information

were reported in *Supplementary file 3*. This score was evaluated for each gene to assess potential correlation between individual gene scores. A random simulation using R software and based on a Chi-square test indicate that the methylation of these genes was independent from one another p<0.01. Kaplan–Meier plots were created using GraphPad Prims8 software. The survival in months was indicated depending whether the score was under vs equal or superior than 2. Survival analysis was performed using a log-rank and the Gehan-Breslow-Wilcoxon test with a p-value<0.001 for each considered significant. Hazard ratio was estimated on R software using survival, survminer, and ggplot2 packages.

## Methylation cluster identification through statistical analysis of the distribution of the hypermethylated genes

Identified methylation clusters highlight regions where the methylation distribution on the chromosome is not random. Maps of the 229 hypermethylated genes were visualized on the Ensembl website using the tool view on karyotype (http://www.ensembl.org). Sex chromosomes were excluded from the analysis. The clusters were defined as a group of at least two methylated genes in close proximity separated by non-methylated genes. The statistical relevance of the number of observed clusters on each chromosome was addressed using bootstrapping. For each simulation, methylated and non-methylated genes were randomly repositioned (shuffled) along each chromosome before recomputing the number of clusters. 1000 simulations were performed to estimate the probability of obtaining the number of observed clusters. All analyses were performed using custom-written scripts implemented in the statistical programming language R (http://cran.r-project.org/). All R-scripts are available from the authors upon request.

## Functional annotation and pathway analysis

The list of hypermethylated genes was imported into QIAGEN's IPA (QIAGEN Redwood City, https://digitalinsights.qiagen.com/). In IPA, hypermethylated genes were mapped to molecular and cellular functions and to networks available in the ingenuity database and then ranked by score or p-value (p<0.05).

## Additional information

### Funding

| Funder | Grant reference number | Author |
|---|---|---|
| Centre National de la Recherche Scientifique | ATIP | Paola B Arimondo |
| Centre National de la Recherche Scientifique | Region Midi Pyrenees - Equipe d'excellence | Paola B Arimondo |
| Centre National de la Recherche Scientifique | Region Midi Pyrenees - FEDER | Paola B Arimondo |
| Fondation InnaBioSante | EpAM | Paola B Arimondo |
| Adelson Medical Research Foundation | | Dave SB Hoon Matias Bustos |

The funders had no role in study design, data collection and interpretation, or the decision to submit the work for publication.

### Author contributions

Arnaud Carrier, Data curation, Investigation, Methodology, Writing – original draft; Cécile Desjobert, Formal analysis, Investigation, Methodology, Writing – review and editing; Loic Ponger, Antoine Daunay, Data curation, Investigation; Laurence Lamant, Matias Bustos, Jorge Torres-Ferreira, Rui Henrique, Luisa Lanfrancone, Audrey Delmas, Gilles Favre, Resources; Carmen Jeronimo, Resources, Data curation; Florence Busato, Data curation, Methodology; Dave SB Hoon, Resources, Data curation, Writing – review and editing; Jorg Tost, Data curation, Methodology, Writing – review and

editing; Chantal Etievant, Conceptualization, Supervision, Methodology, Writing – original draft, Project administration, Writing – review and editing; Joëlle Riond, Conceptualization, Supervision, Investigation, Methodology, Writing – review and editing; Paola B Arimondo, Conceptualization, Formal analysis, Supervision, Funding acquisition, Validation, Investigation, Methodology, Writing – original draft, Project administration, Writing – review and editing

### Author ORCIDs
Arnaud Carrier http://orcid.org/0000-0002-9200-7685
Joëlle Riond http://orcid.org/0000-0002-6281-2376
Paola B Arimondo http://orcid.org/0000-0001-5175-4396

### Ethics
Human subjects: Tumour samples from melanoma patients were obtained from the tumour tissue bank at the Department of Pathology, IUCT-O Toulouse Hospital (France). The study was carried out in accordance with the institutional review board-approved protocols (CRB, AC-2013-1955) and the procedures followed were in accordance with the Helsinki Declaration. This study was approved by the institutional ethics committee of IPO Porto (CES-IPOP-FG13/2016).

### Decision letter and Author response
Decision letter https://doi.org/10.7554/eLife.78587.sa1
Author response https://doi.org/10.7554/eLife.78587.sa2

## Additional files

### Supplementary files
- Supplementary file 1. Ingenuity Pathway Analysis (IPA) software networks results.
- Supplementary file 2. Bisulphite pyrosequencing primer, sequences, and CpG location.
- Supplementary file 3. Survival data and signature score for each patient sample.
- MDAR checklist
- Source code 1. R scripts.

### Data availability
Sequencing data have been deposited in GEO under accession code GSE155856. R-scripts are available in Source Code 1. The datasets supporting the conclusions of this article are included within the article and the following Supplementary files.

The following dataset was generated:

| Author(s) | Year | Dataset title | Dataset URL | Database and Identifier |
|---|---|---|---|---|
| Arimondo PB | 2022 | DNA methylome combined with chromosome cluster-oriented analysis provides an early signature for cutaneous melanoma aggressiveness | http://www.ncbi.nlm.nih.gov/geo/query/acc.cgi?acc=GSE155856 | NCBI Gene Expression Omnibus, GSE155856 |

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
