## [Editor Report]

Predicting if a tumour has aggressive or metastatic characteristics would be of great utility in the clinic as it would help patient management. In this manuscript, Carrier and collaborators derive a signature for melanoma aggressiveness relying on methylated regions of tumour and cell line genomes. The approach the authors take is innovative as it relies on the premise that genes that make cells be more aggressive should be detected across different organisms. In their results, the authors devise a DNA methylation score that correlates with survival and can be potentially useful for patient stratification.

---

## [Decision Letter]

**Decision letter after peer review:**

Thank you for submitting your article "DNA methylome combined with chromosome cluster-oriented analysis provides an early signature for cutaneous melanoma aggressiveness" for consideration by *eLife*. Your article has been reviewed by 3 peer reviewers, including C Daniela Robles-Espinoza as the Reviewing Editor and Reviewer #1, and the evaluation has been overseen by W Kimryn Rathmell as the Senior Editor. The following individual involved in the review of your submission has agreed to reveal their identity: Florian A Karreth (Reviewer #2).

Essential revisions:

1. As some of the cell lines were derived from the same patient but not by the researchers themselves (they were purchased from companies), do they know how different these cell lines are (genetically or epigenetically) purely due to factors such as time of sample collection (i.e. were primaries and metastases taken at the same time, time in tissue culture, etc?) Do the companies provide this information? In the opinion of a reviewer, this is important to consider due to the possibility that more genes may be methylated for other reasons as more time passes, therefore complicating the analysis.

2. Please explain why gene function was one of the means (non-oriented strategy) used for selecting candidate hypermethylated genes. A biomarker does not have to also have a cancer-associated function.

3. How do the hypermethylated genes compare with those that have been identified by other groups that study melanoma metastasis (e.g. these papers PMID 26091043, 32406600)? Do they overlap? Do the genes identified in Carrier et al. constitute a 'methylation group' as identified by these other works? If the signature is as robust as claimed, shouldn't it come up in other studies? A thorough comparison with other findings should be made, especially if the detected signature is being proposed as a clinical biomarker.

4. The robustness of the signature seems to be calculated from the same data from where it was derived (well, more tumours were added but the original from where the signature was derived remains in the dataset). A reviewer points out that for calculating the robustness it would need to be tested on a completely independent set of tumours. In order to claim that it is a robust signature for predicting outcome, and as the authors are proposing to make a kit from it, a higher number of independent tumours should be evaluated. Can the signature be tested in TCGA data? What about the influence of ethnicity or other factors in outcome/methylation, that would maybe influence the clinical applicability of the signature?

5. Other than the listed justification for selecting the 8 genes for validation, was there a unifying rationale? The selection appears very arbitrary.

6. There seems to be a difference between cell lines and patient samples for MYH1, which is hypermethylated in the former but hypomethylated in the latter. Can the authors speculate why that is? Is MYH1 a good marker if the methylation status is different in cells and patient specimens?

7. In line 524, it says "Going forward, a signature could be used to develop a much-needed early diagnostic DNA methylation kit". Do the authors envision that these genes can help in the diagnosis of melanoma? But these were derived from comparing metastasis/aggressive cell lines vs their counterpart – perhaps it would potentially be more useful in predicting metastasis?

8. Why was 1 year chosen to separate short and long survival? Is there a clinical or scientific rationale?

9. In the Discussion, it would be good to add why the authors think these four specific genes came up – what do they do in isolation that seems so critical for tumour progression? Also, are four genes really enough to generate a clinically applicable signature for metastasis/survival? Wouldn't other factors (e.g. mutations, or gene expression) need to be taken into account to predict this phenomenon? Are the 4 methylated genes also differentially expressed and do expression levels correlate with survival? Could the expression of these 4 genes be used to create a score for prognosis?

[Editors' note: further revisions were suggested prior to acceptance, as described below.]

Thank you for resubmitting your work entitled "DNA methylome combined with chromosome cluster-oriented analysis provides an early signature for cutaneous melanoma aggressiveness" for further consideration by *eLife*. Your revised article has been evaluated by W Kimryn Rathmell (Senior Editor) and a Reviewing Editor.

The manuscript has been improved but there are some remaining issues that need to be addressed, as outlined below:

1. Both reviewers agree that it is unclear why the authors state that the methylation status need not be consistent between cell lines and patient samples to be a useful biomarker, and the reviewers consider that further discussion is warranted. The authors state throughout the manuscript and in the response to the reviewers that their prognostic signature relies on 5 CpGs but not on the 4 genes themselves. If so, why were genes with putative functions in cancer prioritized in the first place? Conversely, if the CpG harboring genes have roles in cancer, as the authors have shown for miR-199 and PCDHB15, wouldn't the methylation state (hyper vs hypo in the case of MYH1) be important? In other words, is it valid to simply score for changes in methylation rather than increase vs decrease for a biomarker, and if so, why? This seemingly shifting focus should be better discussed.

2. A reviewer argues that, although they understand the authors' explanation that the identified methylated regions are not all measured in publicly available datasets, the fact remains that the signature has not been validated in a larger dataset. Therefore, they consider it a little premature to claim that the identified methylated regions can be incorporated into a kit to test melanoma progression risk. We recommend that this section in the discussion is softened.

3. If the score for BCL2L10 is the mean of the two CpGs, then isn't it possible that this marker will not score in cases where methylation of one CpG is increased while the other is decreased and they cancel each other out? This should also be discussed.

4. Finally, we ask that some explanations provided for the reviewers also be included in the main text, specifically, the answer to major points #2, #5 (which are not clear from the revised text), #7, and #8.

---

## [Author Response]

Essential revisions:1. As some of the cell lines were derived from the same patient but not by the researchers themselves (they were purchased from companies), do they know how different these cell lines are (genetically or epigenetically) purely due to factors such as time of sample collection (i.e. were primaries and metastases taken at the same time, time in tissue culture, etc?) Do the companies provide this information? In the opinion of a reviewer, this is important to consider due to the possibility that more genes may be methylated for other reasons as more time passes, therefore complicating the analysis.

We agree with the referees and editors that this is an important point to consider. According to the information provided by the American Type Culture Collection, the sampling ages were 55 years for both WM115 and WM266-4 cell lines (established from a female patient), and 54 years for the WM983A and WM983B cell lines (male patient). This suggests that the compared cell lines were established either simultaneously or with a short delay. After reception from ATCC, the cell lines were amplified and rapidly frozen in order to perform experiments with low-passage cells (<20). The same was done with the cell lines M4Be, TW12 and M4BeS2 that are lab strains. Whereas one cannot exclude that some modifications of DNA methylation occurred through the process of establishing and culturing the cells, careful cell culture practices and experimental planning were applied to limit the cell culture process-related divergence of DNA methylation between the cell line to be compared.

To clarify this point, the material and methods section was modified at page 4 and a comment was added in the Results section at page 9.

Noteworthy, the bias induced by the in vitro culture of the cell lines should be minimized by our working hypothesis i.e.; the methylation traits of aggressiveness are independent of the context and common among pairs of cell lines for which the aggressiveness is obtained in different settings.

2. Please explain why gene function was one of the means (non-oriented strategy) used for selecting candidate hypermethylated genes. A biomarker does not have to also have a cancer-associated function.

We thank the referee for raising this point. Indeed, biomarkers do not need to be associated to a cancer function and the 5 CpGs that we have found are on 4 different gene promoters and are not necessarily associated to the expression of the corresponding gene. Nevertheless, in the initial selection strategy, we made the opportunistic choice to restrict the analysis to CpGs in promoters or first exons that are associated to key biological processes to have the possibility to focus in parallel on candidates having a biological function associated to DNA methylation changes in cutaneous melanoma. This analysis allowed us to identify that miR199-3p is silenced by promoter hypermethylation promoting metastasis formation (Dejobert et al. Clin. Epigenetics 2019, doi: 10.1186/s13148-018-0600-2) and that one of the 4 genes that bear the 5 CpGs of the signature plays a role in inhibiting invasion, aggregation and lung metastasis formation in vivo when stably overexpressed in metastatic melanoma cells (manuscript submitted, attached as supplementary file for the referees only). Finally, we have also analyzed the expression level and the impact of the DNA demethylation treatment by 5AzadC on the expression of the 8 genes to assess the reversibility by RT-PCR and Rt-qPCR, but this was only informative and was not used in the selection process.

3. How do the hypermethylated genes compare with those that have been identified by other groups that study melanoma metastasis (e.g. these papers PMID 26091043, 32406600)? Do they overlap? Do the genes identified in Carrier et al. constitute a 'methylation group' as identified by these other works? If the signature is as robust as claimed, shouldn't it come up in other studies? A thorough comparison with other findings should be made, especially if the detected signature is being proposed as a clinical biomarker.

It is difficult to compare our DNA methylation signature with the public data available or other studies because of the selected 5 CpGs only two are present on the 450k Illumina bead chip used in the studies (cg16842280 in MYH1 and cg12067522 in BCL2L10), while the other three are near Illumina probes but are not analyzed by others. We have selected these because the showed the highest methylation changes in the first set of patient samples that we used. In addition, the first paper cited, PMID 260991043, performed genome sequencing looking for mutations and measured expression and protein levels but not DNA methylation profiles. Whereas reference PMID32406600, as discussed in the Discussion section at page 20, analyzed the DNA methylation only by the Illumina 450k array without using another complementary technique, thus the data are not available for all the five CpGs that we selected. However, as suggested by the referees, we have calculated a score for the 2 CpG available in this study (MYH1 and BCL2L10) and we obtain a similar trend with a p=0.01 with the log rank test and non significant with the wilcoxon test. To have a significant score we need the methylation values of the 5 CpGs simultaneously.

**Author response image 1. sa2fig1:** 

4. The robustness of the signature seems to be calculated from the same data from where it was derived (well, more tumours were added but the original from where the signature was derived remains in the dataset). A reviewer points out that for calculating the robustness it would need to be tested on a completely independent set of tumours. In order to claim that it is a robust signature for predicting outcome, and as the authors are proposing to make a kit from it, a higher number of independent tumours should be evaluated. Can the signature be tested in TCGA data? What about the influence of ethnicity or other factors in outcome/methylation, that would maybe influence the clinical applicability of the signature?

We agree with the referee but as described above the public data available do not contain the analysis of all the 5 CpGs that are reported in our study. This has been clarified in the Discussion part at page 19. It is however important to underline that we did proceed in two steps. The first was the selection of the CpGs that showed the most significant changes among a first set of 18 patient samples from primary and metastatic tumors (data not shown).

This highlighted the difference between the primary and metastatic tumors and a clustering of the primary in two subsets that were found related to the delay of survival. Then we analyzed the selected CpGs in a bigger cohort of primary tumors collected from different centers (49 in total) reported in Figure 4 of the manuscript. The application of the signature of 5 CpGs that we have identified as clinical biomarker are beyond the scope of the manuscript, but it is something that we are pursuing. The fact that the signature consists in the methylation level of 5 CpG has the advantage that it can be multiplexed in one assay. However, primary tumors are mainly detected and removed at town dermatologists so it is important to develop an assay that can be readily used. We are investigating several options. Here we report, the following main and original results: (1) the observation that robust DNA methylation traits of aggressiveness are independent of the physiological context; (2) the methodology of combining DNA methylome analysis and chromosome cluster-based analysis that can be applied beyond melanoma, and (3) the identification of the methylation of 5 CpGs (and not genes) that provide a predictive value of the aggressiveness of the melanoma in primary tumors.Concerning the influence of other factors, we analysed the Breslow factor and the combination of the two has a very positive impact on the hazard ratio (Figure 4-Supplement 2). Other additional factors were not analysed because of the limited size of the cohort, however it is important to underline that the samples we collected in centers located in different countries in Europe (France, Portugal and Italy) and in the US (California), and that the signature is robust across these diverse environments.

5. Other than the listed justification for selecting the 8 genes for validation, was there a unifying rationale? The selection appears very arbitrary.

We thank the referees for the comment as we realize that this point needed to be clarified. he choice was not arbitrary. The 8 genes chosen among the 19 common genes obtained from the oriented and non-oriented strategies have the following features: (1) they were distributed on chromosomes with clusters of methylation, (2) they correspond to peaks of hypermethylation and 3/ have a potential function role in cancer formation. For example, we were interested in genes that were not described in melanoma but showed a functional role in other cancers, as the PCDHB and MYH clusters. This has been clarified in the Results section at page 13 before the description of the genes.

6. There seems to be a difference between cell lines and patient samples for MYH1, which is hypermethylated in the former but hypomethylated in the latter. Can the authors speculate why that is? Is MYH1 a good marker if the methylation status is different in cells and patient specimens?

We were looking for the most important changes in the methylation values at single CpGs independently of the decrease or increase. The CpG found in MYH resulted to be correlated to a decrease in methylation rather than an increase in the first set of patient samples and thus we kept it considering this feature. It is important to note that here we did not focus on genes but rather on CpGs and in total five CpGs that are located on four different genes showed the most important DNA methylation changes. This is in agreement with comment #2 that the biomarkers do not need to be associated to cancer functions.

7. In line 524, it says "Going forward, a signature could be used to develop a much-needed early diagnostic DNA methylation kit". Do the authors envision that these genes can help in the diagnosis of melanoma? But these were derived from comparing metastasis/aggressive cell lines vs their counterpart – perhaps it would potentially be more useful in predicting metastasis?

This is exact. We have clarified the sentence in the text that was confusing. The 5 CpGs do not help to diagnose melanoma, they show, in primary tumors, a predictive score of the aggressiveness of the melanoma meaning the survival rate. Thus, the methylation level of the five CpGs combined for example with the Breslow score (Figure 4-Supplement 2) can be used to identify the patients that are at risk when primary tumors are collected and removed.

8. Why was 1 year chosen to separate short and long survival? Is there a clinical or scientific rationale?

One year was chosen because at the time of the sampling and the beginning of the study it was the average overall survival of diagnosticated cutaneous melanoma. Recently it has been extended thanks to immune therapies but it is still not applicable at all cutaneous melanomas. In addition, the delay of one year well distinguished the two group of long and short survival.

9. In the Discussion, it would be good to add why the authors think these four specific genes came up – what do they do in isolation that seems so critical for tumour progression? Also, are four genes really enough to generate a clinically applicable signature for metastasis/survival? Wouldn't other factors (e.g. mutations, or gene expression) need to be taken into account to predict this phenomenon? Are the 4 methylated genes also differentially expressed and do expression levels correlate with survival? Could the expression of these 4 genes be used to create a score for prognosis?

It is important to underline that it is not 4 genes but 5 CpGs located in 4 different genes. The purpose is not to use gene expression – that is not regulated by a single CpGs-, but the difference in methylation levels of these CpGs. For a biomarker, 5 CpGs can be sufficient. Lyskjaer et al. have recently shown that the methylation at 4 CpGs is a prognostic biomarker for osteosarcoma (Lyskjaer et al. Eur J Cancer. 2022 doi: 10.1016/j.ejca.2022.03.002). This has been underlined in the discussion and the reference has been added at page 22.

As suggested by the referees, we have indeed looked at the expression of the genes aiming at identifying potential functional candidates for melanoma aggressiveness. This led to the identification of the PCDHB cluster and in particular PCDHB15 that we have further characterized.

[Editors' note: further revisions were suggested prior to acceptance, as described below.]

The manuscript has been improved but there are some remaining issues that need to be addressed, as outlined below:1. Both reviewers agree that it is unclear why the authors state that the methylation status need not be consistent between cell lines and patient samples to be a useful biomarker, and the reviewers consider that further discussion is warranted. The authors state throughout the manuscript and in the response to the reviewers that their prognostic signature relies on 5 CpGs but not on the 4 genes themselves. If so, why were genes with putative functions in cancer prioritized in the first place? Conversely, if the CpG harboring genes have roles in cancer, as the authors have shown for miR-199 and PCDHB15, wouldn't the methylation state (hyper vs hypo in the case of MYH1) be important? In other words, is it valid to simply score for changes in methylation rather than increase vs decrease for a biomarker, and if so, why? This seemingly shifting focus should be better discussed.

We thank the reviewers for their comment because it indicates that it is not clear in our text. As we wanted to identify specific methylation states monitoring melanoma aggressiveness, we chose not to focus on gene expression, but rather on CpGs that were highly differentially methylated (40%) in aggressive melanoma compared to the non-aggressive counterpart and CpGs that are common in whatever physiological environment the melanoma acquired the aggressive trait (Figure 1 and page 4, lines 107-109 and added text lines 117-120 and page 6 lines 147-149 in the tracked version of the manuscript). DNA methylation of promoters is commonly associated to gene repression as it can shut down genes but it is not the only factor and for example treatment with DNA methylation inhibitors, such as azacitidine and decitabine, alone does not always reactivate gene expression. That is why we underline that we are not looking at the genes and their expression but specifically at the methylation status of certain CpGs. In addition, there is no clear indication that the methylation of one single CpG is regulator of gene expression. But as the referees suggest these methylated CpGs play a role in cancer formation and progression and certainly are a way of monitoring cancer formation and progression, that is on what is based our work. We thus first looked for the genomic distribution of the methylated CpGs and selected CpGs that are highly methylated, are regrouped in clusters and form peak of methylation (Figure 2). This was based on two hypothesis (described in the beginning of the Discussion section at page 13): highly methylated CpG sites correspond to peaks of methylation and could be the start of the spreading of DNA methylation, as shown in the literature, occurring very early in formation of the cancer. We considered that if they are common to melanoma cells that acquired aggressivity in very different physiological context then they would be early markers and important for cancer progression, thus potentially appropriate biomarkers. This oriented strategy, as indicated in figure 1, selected 34 genes. Then we reasoned, as the referees underlined in their comment, that, if this specific methylation is common to different aggressive melanoma, it should play a role in cancer and thus we carried out an IPA analysis (non-oriented strategy, page 7, figure 1) leading to 19 common genes to both strategies and after the analysis of the literature to 8 genes (page 8 lines 197-200 and page 14 lines 347-348). The explanation has been added in the Results section at lines 180-182 page 7. In parallel, this analysis also allowed us to identify the role of miR-199 and PCDHB15 in melanoma aggressiveness, but this role of tumour suppressor genes is not fundamental for the identification of biomarkers and is independent of it. For example, the CpGs of miR-199 were not confirmed for the methylation signature from the beginning of the pipeline process.

What we want to underline is that the signature proposed here is based on DNA methylation not on gene expression and more importantly on a strong and specific difference of DNA methylation level. For the development of potential biomarkers based on DNA methylation it is not necessary that there is correlation with gene expression (Lyskjaer et al. Eur J Cancer. 2022 doi: 10.1016/j.ejca.2022.03.00).

It is not surprising to see DNA methylation differences between patient samples and cells lines as it is also known that the environment, as cell culture, can affect DNA methylation. To increase the robustness of the results, we looked for patient samples from different regions (France, Italy, Portugal and California, as specified in the Materials and methods and added at page 14 line 369) to help identify the CpGs which difference in methylation level was the most robust and independent of the context. Thus, the initial original hypothesis that leverages the heterogeneity of melanoma and the robustness of the DNA methylome together with the multi-step process identified 5 CpGs, which showed the highest and most consistent difference in methylation levels. Finally, the methylation score is possible by the combination of the methylation status of the CpGs but not their role in expression of the corresponding gene. It is this methylation score that enables to classify the patient samples and not one by one or other combinations (Figure 4 and supplements).

We have modified the text to clarify this point, by adding in the results session our reasoning that previously was only described in the introduction and Discussion section. We have also added lines 328 to 333 at page 13 to underline the originality of the results.

2. A reviewer argues that, although they understand the authors' explanation that the identified methylated regions are not all measured in publicly available datasets, the fact remains that the signature has not been validated in a larger dataset. Therefore, they consider it a little premature to claim that the identified methylated regions can be incorporated into a kit to test melanoma progression risk. We recommend that this section in the discussion is softened.

We completely agree with the reviewer and we have modified the text at page 15 lines 370374.

3. If the score for BCL2L10 is the mean of the two CpGs, then isn't it possible that this marker will not score in cases where methylation of one CpG is increased while the other is decreased and they cancel each other out? This should also be discussed.

As underlined by the referee, the two CpGs are indeed close to each other (Figure 3 and supplement dataset1). In all our sampling the methylation level moved in the same way with similar levels for both CpG. This can be seen in the dot plot in Figure 4B in which the two CpG have been reported separately and in the heatmap below of metastasis and primary samples.

Since they move in the same direction and have a comparable difference in methylation between short-survival and long-survival samples, we took the mean value for the methylation score. In case, the two CpGs levels move in different direction it is true that they will cancel each other, but it i will be compensated by the other three CpGs of the signature. This has been added in the text at page 12 lines 289-291.

4. Finally, we ask that some explanations provided for the reviewers also be included in the main text, specifically, the answer to major points #2, #5 (which are not clear from the revised text), #7, and #8.

We have added point #2 at page 6 lines 147-149 and at page 15 lines 379-382, #5 at page 8 lines 197-200, #7 at page 15 lines 370-374 and #8 at page 10 lines 256-259.